# Neutral Position Facilitates Nasotracheal Intubation with a GlideScope Video Laryngoscope: A Randomized Controlled Trial

**DOI:** 10.3390/jcm9030671

**Published:** 2020-03-02

**Authors:** RyungA Kang, Ji Seon Jeong, Justin Sangwook Ko, Jaemyung Ahn, Mi Sook Gwak, Soo Joo Choi, Ji Yun Hwang, Tae Soo Hahm

**Affiliations:** 1Department of Anesthesiology and Pain Medicine, Samsung Medical Center, Sungkyunkwan University School of Medicine, Seoul 06351, Koreajsko@skku.edu (J.S.K.); ms.gwak@samsung.com (M.S.G.); sjoo.choi@samsung.com (S.J.C.); jyun.hwang@samsung.com (J.Y.H.); ts.hahm@samsung.com (T.S.H.); 2Department of Dentistry, Samsung Medical Center, Seoul 06351, Korea; jaemyung.ahn@samsung.com

**Keywords:** anesthesia, intubation, videolaryngoscope

## Abstract

The optimal head position for GlideScope facilitated nasotracheal intubation has not yet been determined. We compared the neutral and sniffing positions to establish the degree of intubation difficulty. A total of 88 ASA I-II patients requiring nasotracheal intubation for elective dental surgery with normal airways were divided into two groups according to head position, neutral position (group N), and sniffing position (group S). The primary outcome was the degree of intubation difficulty according to the Intubation Difficulty Scale (IDS): Easy (IDS = 0), mildly difficult (IDS = 1 to 4), and moderately to severely difficult (IDS ≥ 5). Eighty-seven patients completed the study and their data were analyzed. The degree of intubation difficulty was significantly different between the two groups (*p* = 0.004). The frequency of difficult intubation (IDS > 0) was 12 (27.9%) in group N and 28 (63.6%) in group S (difference in proportion, 35.7%; 95% confidence interval [CI], 14.8 to 52.6%; *p* = 0.001). The neutral position facilitates nasotracheal intubation with GlideScope by aligning the nasotracheal tube and the glottis inlet more accurately than the sniffing position.

## 1. Introduction

The GlideScope video laryngoscope (Verathon, Bothell, WA, USA), used for intubation, can easily achieve a good laryngeal view [1]. A good laryngeal view is important for tracheal intubation, but this alone does not guarantee ease of intubation. Clinicians frequently experience technical difficulties for placement of the tracheal tube during tracheal intubation using GlideScope with a hyperangulated curvature blade (60-degree) [2]. The cause of this difficulty may be due to malalignment of the tip of the tracheal tube with the glottic opening [3]. To overcome this malalignment, changes to head and neck positions can be suggested, as these are known to affect the success of intubation [4]. Nasotracheal intubation is often indicated in oral and maxillofacial surgery. Compared to a direct laryngoscope, nasotracheal intubation with the GlideScope makes intubation faster and easier and plays a clear role in routine nasotracheal intubation [5]. Previous studies suggest that a neutral head position further augmented blind nasotracheal intubation with or without the cuff inflation technique when using a lightwand device [6,7]. However, the optimal head position for GlideScope-facilitated nasotracheal intubation has not yet been determined. We aimed to evaluate whether the neutral head position facilitates nasotracheal intubation by comparing it with the sniffing position when using a GlideScope.

## 2. Materials and Methods

This study was approved by the Samsung Medical Center Institutional Review Board, Seoul, Korea (identifier SMC 2018-03-120-003) and prospectively registered at the Clinical Trial Registry of Korea (http://cris.nih.go.kr; identifier KCT0002924, registration date: June 14, 2018, presenting author: Ji Seon Jeong). This manuscript adheres to the applicable CONSORT guidelines [8]. Written informed consent was obtained from all participants. Eighty-eight adult patients with normal airways, scheduled for elective dental surgery under general anesthesia that required nasotracheal intubation, were enrolled between May 2018 and January 2019 at Samsung Medical Center, Seoul, Korea. Eligible patients were identified from the surgeon’s operating list and contacted by phone the day prior to their surgery to inform them of the study protocol (RAK). Exclusion criteria included a past history (cervical spine injury or history of upper airway surgery) or clinical signs of a potentially difficult airway (modified Mallampati score of 3 or 4 [9], mouth opening <30 mm, thyromental distance <65 mm, limited neck movement <80° by the Wilson test) [10,11], bleeding tendency, body mass index >30 kg·m^2^, or an American Society of Anesthesiologists (ASA) physical status higher than III. 

On the day of surgery, the patient arrived at the outpatient surgery center 2 hours before surgery. A single independent investigator (JYH) who was blinded to the study details visited preoperative waiting area and airway-associated examination including mouth opening, modified Mallampati score, thyromental distance, and neck movement were performed. During the preoperative visit, patients asked the questions and re-considered their participation in the study. Patients were randomly assigned to one of two groups in a 1:1 ratio using a computer-generated block randomized tool (www.randomizer.org) by a member of Samsung Medical Center who was not otherwise involved in the study: The head in neutral position group (group N) or the sniffing position group (group S). Allocation of patients to each study group was concealed in an opaque envelope, which was opened by the practitioner before the general anesthesia. The nasotracheal intubations were performed by one anesthesiologist (JSJ) who had used a GlideScope on at least 200 occasions. All other research personnel, outcome assessors, the patients, and caregivers were blinded to group allocation. 

In the operating room, patients were positioned supine with the head and neck supported on pillows (5 cm), to be as close to a neutral position as possible within their comfort range. All patients underwent standardized general anesthesia induced with intravenous propofol (1.5–2 mg·kg^−1^), rocuronium (0.6–0.8 mg·kg^−1^), and remifentanil 0.01–0.1 µg·kg^−1^·min^−1^. After adequate muscle relaxation, the randomized head position was then applied to patients, and the specified data collection was started. The neutral head position was obtained by lying flat without a pillow (Figure 1A). The sniffing position was obtained by placement of several firm cushions under the head of the patient to make neck flexion with extension of atlanto-occipital joint, confirming that the external auditory meatus and sternal notch plane were horizontal and aligned (Figure 1B) [4]. 

All nasotracheal intubations were performed with GlideScope^®^ AVL single-use blade and recorded on a GlideScope^®^ video monitor (Verathon, Bothell, WA, USA). Nasotracheal intubation was performed with nasal Ring-Adair-Elwin tracheal tubes (Mallinckrodt Medical, Covidien, Dublin, Ireland) with an internal diameter of 7.0 and 6.5 mm for males and females, respectively. The nasotracheal tubes were placed into sterile saline maintained at 45 °C and then lubricated with water-soluble gel. The nasotracheal tube was inserted through the more appropriate nostril to the oropharyngeal region. The GlideScope, using an appropriate blade following the recommendation of the manufacturer, was introduced into the mouth, and the modified Cormack and Lehane grade was assessed [12], and then the nasotracheal tube was directed from the pharyngeal region to the vocal cord. Optimizing maneuvers such as external laryngeal pressure or use of Magill forceps were applied if needed. A failed intubation was defined when the trachea was not intubated or when the time required for intubation was longer than 180 s. After the airway was secured, anesthesia was maintained with sevoflurane in a 1:1 oxygen/air mixture. Thirty minutes before the end of surgery and irrespective of group allocation, fentanyl (1.0 µg·kg^−1^) was administered for postoperative pain control. After surgery, patients were transferred to the post-anesthesia care unit. 

The primary outcome was the rate of difficulty nasotracheal intubation, which was assessed using the modified Intubation Difficulty Scale (IDS) developed by Adnet et al. [13]. Specifically, we defined three groups of patients according to IDS value: Easy (IDS score = 0), mildly difficult (IDS score = 1 to 4), and moderately to severely difficult (IDS score ≥5). Of these, difficult intubation was defined with an IDS score >0. The IDS describes the difficulty of intubation based on the seven parameters: Number of advancement attempts of the tracheal tube >1 (N1; every additional attempt add 1 point); number of additional operators (N2; each additional operator add 1 point); number of alternative intubation techniques including change of materials (tube type); change in approach (from nasotracheal to orotracheal) or use of another technique (cuff inflation technique) (N3); laryngeal view as defined by the modified Cormack and Lehane grade (N4; grade I = 0, full view of the glottis; grade II = 1, only the posterior commissure of the glottis could be seen; grade III = 2, only the epiglottis could be seen; and grade IV = 3, even the epiglottis could not be seen); lifting force required during GlideScope use (N5; 0 = normal, 1 = increased); need to apply external laryngeal pressure to improve glottis positioning (N6; 0 = not applied, 1 = external laryngeal pressure was used); and aid technique in intubation (N7; 0 = not applied, 1 = head extension, 2 = use of Magill forceps). The IDS score is the sum of N1 through N7 and was measured by another observer in the room during intubation. Ideal intubating conditions yield an IDS score of 0, whereas progressively more difficult tracheal intubations result in higher scores. 

Secondary outcomes were time to successful intubation (time interval from insertion of the GlideScope blade into the mouth to when the tip of the nasotracheal tube passed the vocal cords), use of optimization maneuvers, alignment of the tip of the tracheal tube and glottic inlet, laryngeal views using the modified Cormack and Lehane grading system, hypoxic event (SpO2 < 90%) during intubation, and occurrence of major (dental or other airway trauma) or minor complications (visible trauma to lip or oral mucosa or blood on the GlideScope blade). Alignment of the nasotracheal tube tip with the glottis inlet was classified according to GlideScope view and divided into four zones (Figure 2): ‘A’ designates the tip of the nasotracheal tube positioned toward the center of the glottis inlet, ‘B’ designates the tip of the nasotracheal tube positioned between the lower third of the posterior laryngeal wall and corniculate cartilage, ‘C’ designates the tip of the nasotracheal tube positioned around the corniculate cartilage, and ‘D’ designates the tip of the nasotracheal tube positioned between the posterior laryngeal wall and esophagus. If the nasotracheal tube tip was located in the A or B zone, we considered the alignment to be correct. If the nasotracheal tip was located in C or D zone, we noted malalignment. 

If malalignment was present, external laryngeal pressure to larynx or Magill forceps was sequentially applied to facilitate nasotracheal intubation. The severity of a postoperative sore throat was assessed by a study-blinded nurse in the post-anesthetic care unit using a numeric rating scale of 11 points (where 0 = no pain and 10 = severe pain). 

The sample size was determined based on our preliminary (unpublished) study of from 30 patients with a type I error of 0.05 and a power of 80%. This suggested that a minimum of 80 patients would be required to detect a difference of 30% versus 5% difficulty intubation (IDS score > 0) rates in nasotracheal intubation using the GlideScope. To account for 10% dropout, we aimed to enroll a total of 88 patients (*n* = 44, each group). Data are presented as mean (standard deviation, SD), median (interquartile range, IQR), or number (%) as appropriate. Continuous variables were compared using the t-test or Mann-Whitney U test, and the Kolmogorov–Smirnov test was used to explore normality. Categorical variables were analyzed using the chi-square test or Fisher’s exact test, as appropriate. SPSS version 25.0 (SSPS Inc., Chicago, IL, USA) was used for data analysis. A Bonferroni correction was used for multiple comparisons. A *p* value < 0.05 was considered statistically significant. The datasets generated during the current study are available from corresponding author on reasonable request.

## 3. Results

### 3.1. Study Participants

From May 2018 to January 2019, we assessed 98 patients for eligibility. Of these, 10 patients who did not meet the inclusion criteria were excluded, leaving 88 patients (*n* = 44, each group) enrolled in the study (Figure 3). 

One patient in group N was excluded from analysis because unintended blind intubation occurred during tube advancement. Finally, a total of 87 patients completed the study (group N, *n* = 43; group S, *n* = 44), and their data were analyzed. There were no differences in baseline characteristics or airway data of the two groups (Table 1).

### 3.2. Primary Outcome

Intubation parameters are shown in Table 2. 

The degree of intubation difficulty was significantly different between the two groups (*p* = 0.004). The frequency of difficult intubation (IDS > 0) was 12 (27.9%) in group N and 28 (63.6%) in group S (difference in proportion, 35.7%; 95% confidence interval [CI], 14.8 to 52.6%; *p* = 0.001). All patients underwent successful nasotracheal intubation using the GlideScope within three attempts. However, the success rate of tracheal tube advancement at the first attempt was significantly higher in group N compared to group S (95.3% vs. 61.4%; difference in proportion, 33.9%; 95% CI, 17.2 to 49.1%; *p* < 0.001). 

### 3.3. Secondary Outcomes

The frequency of correct alignment between the nasotracheal tube and the glottis inlet was 39 (90.7%) in group N and 24 (54.5%) in group S (difference in proportion, 36.2%; 95% CI, 15.6 to 49.5%; *p* < 0.001). Eight (18.6%) patients in group N and 22 (50.0%) patients in group S required optimization maneuvers for nasotracheal tube advancement (difference in proportion, 31.4%; 95% CI, 11.5 to 48.1%; *p* = 0.003). Of these, two and five patients in groups N and S, respectively, required the use of Magill forceps. There was no hypoxic event in either group (Table 2). No other major complications, such as dental, pharyngeal, tracheal, or laryngeal injury, were found.

## 4. Discussion

In this randomized controlled trial, we demonstrated that the neutral head position facilitates nasotracheal intubation with GlideScope compared to the sniffing position. In addition, the neutral head position provided a favorable condition for tube placement by properly aligning the nasotracheal tube with the glottis inlet. In particular, there was a 34% higher frequency of correct alignment with the glottis inlet in the neutral head position than the sniffing position. As a result, the neutral head position increased the success rate of the first attempt and shortened the time to successful intubation. 

Generally, three challenging sequential steps are needed for successful tracheal intubation [14], achieving a laryngeal view, delivering the tube tip to the glottic opening, and advancing the tracheal tube into the trachea [15]. In Macintosh laryngoscopy, achieving a good laryngeal view is the most important step for successful tracheal intubation. To obtain a good laryngeal view, the three oral, pharyngeal, and tracheal anatomical axes must be aligned [16]. However, when using the GlideScope, it is not necessary to align the three anatomical axes because the instrument was designed to provide a view of the glottis without alignment of the oral, pharyngeal, and tracheal axes [1]. The sniffing position is useful in aligning the three anatomical axes [17] and can be achieved by elevation of the occiput and extension of the head at the atlanto-occipital joint. However, when using the GlideScope, the sniffing position may not be needed for successful intubation. Instead, alignment of the tracheal tube with the glottis inlet may be a more important step for successful tracheal intubation, because some clinicians frequently encounter technical difficulties in advancement of the tracheal tube during tracheal intubation with GlideScope [14,15]. Therefore, an effort to align the tracheal tube with the glottis inlet is needed when using the GlideScope. 

Head and neck positions can be associated with the increasing success rate of intubation [18,19]. A previous study also suggested that ease of nasotracheal intubation is further augmented by placing the patient’s head in a neutral position [6,7]. A possible mechanism by which the neutral position provides favorable conditions during nasotracheal intubation might be ascribed to good alignment of the pharyngo-tracheal axis. A previous study of un-anesthetized volunteers who underwent magnetic resonance imaging of the head and neck region showed that the nasopharynx was located below the oropharynx in the neutral position but above the oropharynx in the sniffing position [20]. In the neutral position, the tip of the curve-shaped tracheal tube will be directed at the glottis inlet because the oropharynx is higher than the nasopharynx. Conversely, in the sniffing position, the tip of the curve-shaped tracheal tube will be directed at the esophagus because the oropharynx is lower than the nasopharynx. Our results showed that the frequency of correct alignment between the tip of the tracheal tube and the glottis inlet was about 34% higher in the neutral position than in the sniffing position. Another factor that could affect the pharyngo-laryngeal axis is the lifting force of the laryngoscope during the intubation process. The GlideScope video laryngoscope minimizes pharyngo-tracheal axis changes during intubation because its lifting force is less than that of the Macintosh laryngoscope [21]. Therefore, when using the Macintosh direct laryngoscope during nasal intubation, Magill forceps are often used because of the malalignment between the end of the tracheal tube and the glottic inlet [22]. Consequently, when using the GlideScope combined with neutral head position, it is likely that the long axis of the tracheal tube will be more aligned with the trachea, so the tracheal tube can be inserted with greater ease [23]. Based on our findings, the neutral head position can achieve more accurate alignment with the glottis inlet compared to the sniffing position.

This study has several limitations. First, the practitioner who performed the nasotracheal intubations could not be blinded to the assigned head position. However, we assessed the difficulty or ease of intubation using the IDS score by another observer, which is a widely used scoring system for intubation difficulty and allows an objective and comprehensive assessment of the difficulty during intubation [13]. Second, all nasotracheal intubations were performed by one experienced anesthesiologist. Different results might be expected in the hands of beginners. However, since the use of the GlideScope has a short learning curve [24], the bias from the practitioner can be minimized. Third, we included fairly homogenous patients with apparently normal airways. We wanted to confirm the effects of head position first for patients with normal airways. Therefore, our results cannot extrapolate to patients with expected difficult airways. Recent meta-analysis has shown that video laryngoscope is particularly beneficial for patients with difficult airways [25]. Therefore, it may be applied to patients with difficult airways or to other type of video laryngoscope. Future studies are needed to confirm our finding for patients with expected difficult airways. 

## 5. Conclusions

In conclusion, the neutral position facilitates nasotracheal intubation with the GlideScope by aligning the nasotracheal tube and the glottis inlet more accurately than the sniffing position. In addition, the neutral position may be recommended as the initial head position for nasotracheal intubation with the GlideScope.

## Figures and Tables

**Figure 1 jcm-09-00671-f001:**
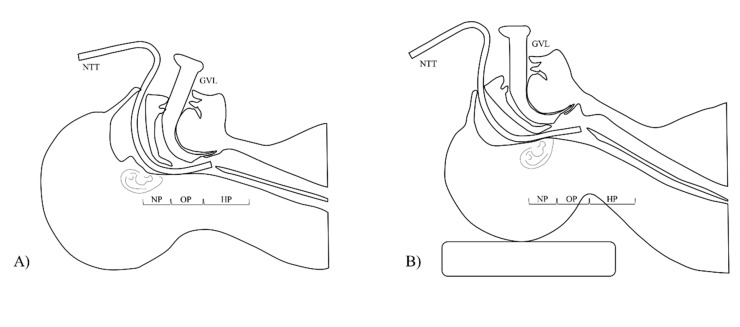
Head positions for intubation. (**A**) Neutral position and (**B**) sniffing position. GVL, GlideScope videolaryngoscope; HP, hypopharynx; NP, nasopharynx; NTT, nasotracheal tube; OP, oropharynx.

**Figure 2 jcm-09-00671-f002:**
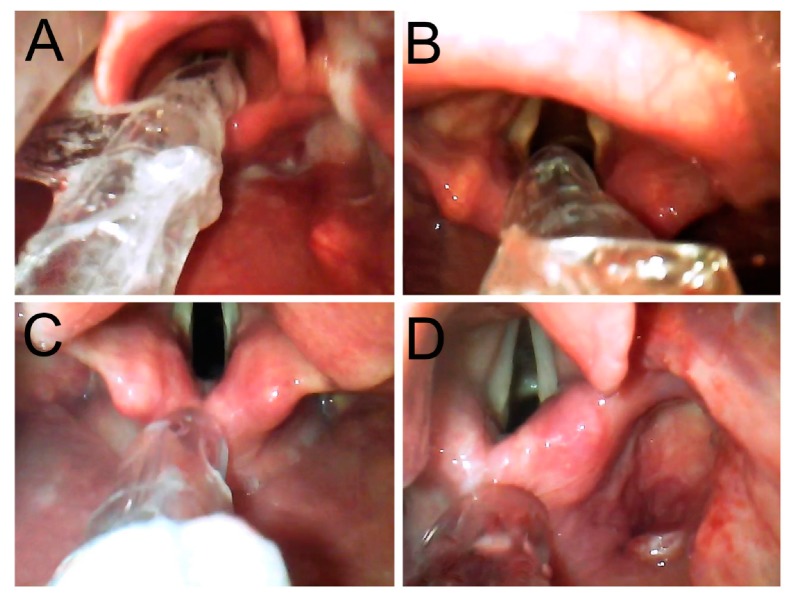
Alignment of nasotracheal tube tip with glottis inlet. The tip of the nasotracheal tube was positioned (**A**) at the center of the glottis inlet, (**B**) between the lower third of the posterior laryngeal wall and corniculate cartilage, (**C**) around the corniculate cartilage, and (**D**) between the posterior laryngeal wall and esophagus.

**Figure 3 jcm-09-00671-f003:**
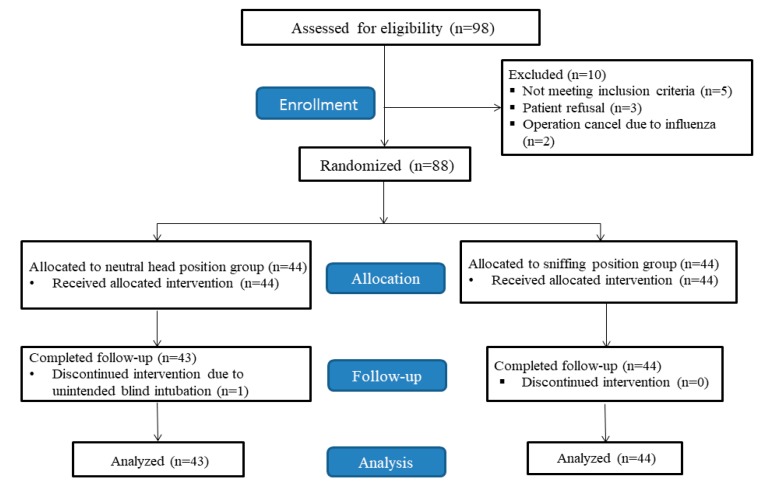
Consolidated Standards of Reporting Trials (CONSORT) diagram.

**Table 1 jcm-09-00671-t001:** Baseline characteristics and preoperative airway assessment.

Parameter	Group N (*n* = 43)	Group S (*n* = 44)
Age (years)	31 (24–56)	33 (23–52)
Sex (male/female)	20/23	24/20
Body mass index (kg/m^2^)	22.8 ± 3.0	23.5 ± 3.1
ASA status (I/II)	20/23	23/21
Modified Mallampati score (I/II)	7/36	4/40
Mouth opening (cm)	3.6 ± 0.3	3.7 ± 0.2
Thyromental distance (cm)	6.9 ± 0.1	6.9 ± 0.1
Nostril (right/left)	31/12	24/20

Values are mean ± standard deviations, median (interquartile range), or number as appropriate.

**Table 2 jcm-09-00671-t002:** Intubation parameters.

Parameters	Group N (*n* = 43)	Group S (*n* = 44)	*p*
Intubation success, n	43	44	
Success of the tracheal tube advancement			
First/Second/Third attempts	41/1/1	27/11/6	0.002
Degree of intubation difficulty (easy vs. mildly vs. moderately and severely)	31/10/2	16/22/6	0.004
Easy (IDS = 0), n	31	16	0.001 *
Mildly difficult (IDS = 1 to 4), n	10	22	0.014 *
Moderately and severely difficult (IDS ≥ 5), n	2	6	0.266 *
^†^ Glottic view (I/II/III/IV)	36/7/0/0	29/15/0/0	0.084
Alignment of the tip of the tracheal tube and glottis inlet			<0.001
Correct alignment (A/B), n	32/7	12/12	
Malalignment (C/D), n	2/2	12/8	
^‡^ Need for optimizing maneuver, n	8	22	0.003
Time to successful intubation, seconds	10 (8–13)	13 (10–20)	0.003
Postoperative sore throat, n	13	11	0.810
Complications			
Mucosal bleeding, n	2	4	0.676
Hypoxia during intubation, n	0	0	N/A

Values are mean ± standard deviations, median (interquartile range), or number as appropriate. N/A, not applicable. * The *p* value is the result of the Bonferroni correction. ^†^ Glottic view was assessed by Cormack and Lehane grade (grade I, full view of the glottis could be obtained; grade II, only the posterior commissure of the glottis could be seen; grade III, only the epiglottis could be seen; and grade IV, even the epiglottis could not be seen). ^‡^ Optimizing maneuver includes external laryngeal pressure or Magill forceps use.

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
