# Peer review of "Neutral Position Facilitates Nasotracheal Intubation with a GlideScope Video Laryngoscope: A Randomized Controlled Trial"

_jcm, 2020, doi:10.3390/jcm9030671_

Round 1

Reviewer 1 Report

This is a prospective randomized study assessing two different positions during nasotracheal intubation with the Glidescope in oral/maxillofacial surgeries.

Major comments:

This study actually shows that an experienced one operator can intubate patients without any predictor of difficult airways in neutral position. It probably does not matter if one intubates patients without difficult airway prediction with a MacIntosh blade or with  any videolaryngoscope. Differences come in difficult airway where the positive role of videolaryngoscope has been confirmed (Jiang et al., J Clin Anesth 2019).

Minor comments:

p.1, l.30 - type of the Glidescope should be described in here

p.2, l.44 - when the patients received the "Study Information Leaflet". Did they heve any time window to ask the questions and consider their participation in the study?

p.2, l.50 - why the authors excluded all patients with any predictors of difficult airway? More beneficial would be to enroll only patients with potentially difficult airways.

p.7, l.222 - why only one experineced operator was chosen for this study?

You stated that GlideScope has a short learning curve - this should be supported by some references. 

Author Response

Response to Reviewer 1 Comments

This is a prospective randomized study assessing two different positions during nasotracheal intubation with the Glidescope in oral/maxillofacial surgeries.

# Major comments:

Point 1. This study actually shows that an experienced one operator can intubate patients without any predictor of difficult airways in neutral position. It probably does not matter if one intubates patients without difficult airway prediction with a MacIntosh blade or with any videolaryngoscope. Differences come in difficult airway where the positive role of videolaryngoscope has been confirmed (Jiang et al., J Clin Anesth 2019).

Response 1. We fully agree with your comments. It may be more beneficial to enroll patients with difficult airways. However, in the conceptualization of this study, we wanted to confirm the effects of head position first in patients with normal airways. In addition, it is hard to enroll only patients with difficult airways in our clinical practice because our patients are relatively slim. Moreover, for patients with expected difficult airways, fiberoptic bronchoscopy is still the first choice for airway management. We have cited the Jiang et al. study (reference number 25) and have revised the manuscript in the limitation section as follows; We wanted to confirm the effects of head position first for patients with normal airways. Therefore, our results cannot extrapolate to patients with expected difficult airway. Recent meta-analysis has shown that video laryngoscope is particularly beneficial for patients with difficult airways [25]. Therefore, it may be applied to patients with difficult airways or to other type of video laryngoscope.”

# Minor comments:

Point1. l.30 - type of the Glidescope should be described in here.

Response 1. Thank you for your comments. We have followed the reviewer’s comments as follows; “Clinicians frequently experience technical difficulties for placement of the tracheal tube during tracheal intubation with GlideScope with hyperangulated curvature blade (60-degree) [2].”

In addition, we have revised the Methods section as follows; “All nasotracheal intubations were performed with GlideScope® AVL single-use blade and recorded on a GlideScope® video monitor.”

Point 2. l.44 - when the patients received the "Study Information Leaflet". Did they have any time window to ask the questions and consider their participation in the study?

Response 2. Thank you for your comments. Already mentioned in the Methods section, eligible patients contacted by phone the day prior to their surgery to inform them of the study protocol. Then, patients arrived at the outpatient surgery center 2 hours before surgery. One investigator visited preoperative waiting area and examined airway-associated with parameters. During this waiting time, patients asked the questions and re-considered their participation in the study. We have revised the manuscript in chronological order to increase the clarity (page 2, line 49-55).

Point 3. p.2, l.50 - why the authors excluded all patients with any predictors of difficult airway? More beneficial would be to enroll only patients with potentially difficult airways.

Response 3. Thank you for your valuable comments. We also agree with your comments that it is more beneficial to enroll only patients with potentially difficult airways. However, we wanted to apply this method first for patients with expected normal airways. We think this method can also be applied to patients with expected difficult airways. This point was also raised by reviewer 2. Thus, we have added this point in the Discussion as follows; We wanted to confirm the effects of head position first for patients with normal airways. Therefore, our results cannot extrapolate to patients with expected difficult airway. Recent meta-analysis has shown that video laryngoscope is particularly beneficial for patients with difficult airways [25]. Therefore, it may be applied to patients with difficult airways or to other type of video laryngoscope.”

Point 4. p.7, l.222 - why only one experienced operator was chosen for this study?

Response 4. Thank you for your comments. During the conceptualization this study, we wanted to minimize the bias from the practitioner but this is the major limitation of our study. We have already mentioned it in the limitation, and if several practitioners had done it, our findings would have been further strengthened. Further studies are needed to achieve generality.

Point 5. You stated that GlideScope has a short learning curve - this should be supported by some references. 

Response 5. Thank you for your comments. To support this sentence, we have cited reference number 24.(Rai, Dering, and Verghese 2005) Previous studies looking at intubation times with devices to aid tracheal intubation have suggested a learning experience of 20-25 cases (Halligan and Charters 2003) but, only 8 cases were required when using GlideScope.

[Reference]

Halligan, M., and P. Charters. 2003. 'A clinical evaluation of the Bonfils Intubation Fibrescope', Anaesthesia, 58: 1087-91.

Rai, M. R., A. Dering, and C. Verghese. 2005. 'The Glidescope system: a clinical assessment of performance', Anaesthesia, 60: 60-4.

------------------------------The end------------------------
We hope the revised manuscript will better meet the requirements of your journal for publication. We thank the Editors and the Reviewers of Journal of clinical medicine once again for the constructive review of our paper.

With regards,

Dr. Ji Seon Jeong

Ji Seon Jeong, M.D., Ph. D., Associate professor, Department of Anesthesiology and Pain Medicine, Samsung Medical Center, 81 Irwon ro, Gangnam gu, Seoul 06351, Korea. Tel.: +82-2-3410-2463; Fax: +82-2-3410-0361; E-mail: [email protected]

Reviewer 2 Report

Kang et al present a randomized study examining optimal head positioning for elective nasotracheal intubation using video laryngoscopy (Glidescope). In this well done study, their findings are consistent with prior literature favoring neutral head position for nasotracheal intubation.

I. Introduction

Consider providing a sentence or two regarding the utility of video vs direct laryngoscopy in nasotracheal intubation for maxillofacial surgery for readers who are more familiar with oral intubation. Is one method preferred over the other?

II. Materials and Methods

Lines 71-73: Please clarify the sniffing position used for your study. There may be some variability in the techniques used to achieve this position. Some practitioners stress the atlanto-occipital extension, but this is not depicted in figure 1B. Without the extension, the position depicted has elements of a ramp position.

Line 93: The primary outcome may need to be more specifically defined here. Is it the rate of moderate/difficult intubations between group N and group S. This is suggested by the power analysis which outlines an expected difference in moderate/difficult intubation rates of 5% vs 30%, respectively. This difference was ultimately not statistically significant (p=0.26) owing to a lower than expected rate of moderate/difficult intubation in group S (only 16% compared to the expected rate of 30%). If this is what the study was powered to detect, it was ultimately a negative study, but the results were presented slightly differently. Was the difference in intubation difficulty (p=0.004) expressed as the difference in mean IDS score between groups?

Author Response

Response to Reviewer 2 Comments

Kang et al present a randomized study examining optimal head positioning for elective nasotracheal intubation using video laryngoscopy (Glidescope). In this well done study, their findings are consistent with prior literature favoring neutral head position for nasotracheal intubation.

Point 1. I. Introduction

Consider providing a sentence or two regarding the utility of video vs direct laryngoscopy in nasotracheal intubation for maxillofacial surgery for readers who are more familiar with oral intubation. Is one method preferred over the other?

Response1. Thank you for your valuable comments. We have added a sentence in the introduction as follows;Compared to direct laryngoscope, nasotracheal intubation with the GlideScope makes intubation faster and easier and plays a clear role in routine nasotracheal intubation.(Jones et al. 2008)

  1. Jones, P. M., K. P. Armstrong, P. M. Armstrong, R. A. Cherry, C. C. Harle, J. Hoogstra, and T. P. Turkstra. 2008. 'A comparison of glidescope videolaryngoscopy to direct laryngoscopy for nasotracheal intubation', Anesth Analg, 107: 144-8.

Point 2. II. Materials and Methods

Lines 71-73: Please clarify the sniffing position used for your study. There may be some variability in the techniques used to achieve this position. Some practitioners stress the atlanto-occipital extension, but this is not depicted in figure 1B. Without the extension, the position depicted has elements of a ramp position.

Response 2: Thank you for your valuable comments. We have revised the Figure 1 B according to your comments. We have reflected the atlanto-occipital extension in the Figure 1B.

Point 3. Line 93: The primary outcome may need to be more specifically defined here. Is it the rate of moderate/difficult intubations between group N and group S. This is suggested by the power analysis which outlines an expected difference in moderate/difficult intubation rates of 5% vs 30%, respectively. This difference was ultimately not statistically significant (p=0.26) owing to a lower than expected rate of moderate/difficult intubation in group S (only 16% compared to the expected rate of 30%). If this is what the study was powered to detect, it was ultimately a negative study, but the results were presented slightly differently. Was the difference in intubation difficulty (p=0.004) expressed as the difference in mean IDS score between groups?

Response 3. Thank you for your valuable comments. We agree with your comments. We have revised the definition of primary outcome to be more specifically in the Methods section as follows; The primary outcome was the rate of difficulty intubation in nasotracheal intubation, which was assessed using the modified Intubation Difficulty Scale (IDS) developed by Adnet et al [13]. Specifically, we defined three groups of patients according to IDS value: easy (IDS score=0), mildly difficult (IDS score=1 to 4), and moderately to severely difficult (IDS score ≥5). Of these, difficulty intubation was defined with an IDS score >0.”

In addition, we made a mistake for the description of sample size. The sample size was determined to detect a difference of 30% vs. 5% difficult intubation rates, not a moderately and severely difficult intubation. We have revised it appropriately. The p-value for degree of intubation difficulty presented in Table 2 is the results of chi-square test comparing three groups of patients (easy, mildly, and moderately and severely). We have added the results of the subgroup analysis and have explained the adjusted p-values at the bottom of Table 2.

----------------------------------The end-----------------------------------------------
We hope the revised manuscript will better meet the requirements of your journal for publication. We thank the Editors and the Reviewers of Journal of clinical medicine once again for the constructive review of our paper.

With regards,

Dr. Ji Seon Jeong

Ji Seon Jeong, M.D., Ph. D., Associate professor, Department of Anesthesiology and Pain Medicine, Samsung Medical Center, 81 Irwon ro, Gangnam gu, Seoul 06351, Korea. Tel.: +82-2-3410-2463; Fax: +82-2-3410-0361; E-mail: [email protected]

Reviewer 3 Report

This report clealy presents the appropriate position for the nasotracheal tube insertion. I agree with this result and conclusion. I think this principle of head position for nasotracheal intubation can be applied for other intubation devices. Please add the comment for this point. Otherwise this article should be congraturated. 

In the legend of Figure2,  Uppercase of A,B,C,D in the photos are used. Lower cases are used in the legend. Please correct them. 

Author Response

Response to Reviewer 3 Comments

Point 1. This report clearly presents the appropriate position for the nasotracheal tube insertion. I agree with this result and conclusion. I think this principle of head position for nasotracheal intubation can be applied for other intubation devices. Please add the comment for this point. Otherwise this article should be congratulated. 

Response 1. Thank you for your valuable comments. We have followed your suggestion and have revised the manuscript as follows; “Therefore, it may be applied to patients with difficult airways or to other type of video laryngoscope.

Point 2. In the legend of Figure2, Uppercase of A, B,C,D in the photos are used. Lower cases are used in the legend. Please correct them. 

Response 2. Thank you for your comments. We revised the legend of Figure with uppercases.

------------The end-----------------------------------------------
We hope the revised manuscript will better meet the requirements of your journal for publication. We thank the Editors and the Reviewers of Journal of clinical medicine once again for the constructive review of our paper.

With regards,

Dr. Ji Seon Jeong

Ji Seon Jeong, M.D., Ph. D., Associate professor, Department of Anesthesiology and Pain Medicine, Samsung Medical Center, 81 Irwon ro, Gangnam gu, Seoul 06351, Korea. Tel.: +82-2-3410-2463; Fax: +82-2-3410-0361; E-mail: [email protected]
